# Wave-Particle Duality Relation with a Quantum Which-Path Detector

**DOI:** 10.3390/e23010122

**Published:** 2021-01-18

**Authors:** Dongyang Wang, Junjie Wu, Jiangfang Ding, Yingwen Liu, Anqi Huang, Xuejun Yang

**Affiliations:** Institute for Quantum Information & State Key Laboratory of High Performance Computing, College of Computer Science and Technology, National University of Defense Technology, Changsha 410073, China; dongyangwang@quanta.org.cn (D.W.); jiangfangding@quanta.org.cn (J.D.); yingwenliu@quanta.org.cn (Y.L.); anqihuang@quanta.org.cn (A.H.); xjyang@nudt.edu.cn (X.Y.)

**Keywords:** wave-particle duality, quantum coherence, path distinguishability, polarization- entangled photons

## Abstract

According to the relevant theories on duality relation, the summation of the extractable information of a quanton’s wave and particle properties, which are characterized by interference visibility *V* and path distinguishability *D*, respectively, is limited. However, this relation is violated upon quantum superposition between the wave-state and particle-state of the quanton, which is caused by the quantum beamsplitter (QBS). Along another line, recent studies have considered quantum coherence *C* in the $l_{1}$-norm measure as a candidate for the wave property. In this study, we propose an interferometer with a quantum which-path detector (QWPD) and examine the generalized duality relation based on *C*. We find that this relationship still holds under such a circumstance, but the interference between these two properties causes the full-particle property to be observed when the QWPD system is partially present. Using a pair of polarization-entangled photons, we experimentally verify our analysis in the two-path case. This study extends the duality relation between coherence and path information to the quantum case and reveals the effect of quantum superposition on the duality relation.

## 1. Introduction

Bohr’s principle of complementarity [1,2] provides the essence of quantum mechanics. It states that a quantum system (i.e., quanton [3]) may possess multiple properties that cannot be observed simultaneously. Wave-particle duality [4] is a well-known example. For a quanton in an interferometer, its path information and interference visibility are incompatible, such that the appearance of one will suppress the other. The general case wherein we obtain incomplete path information and interference visibility is given by the duality relation proposed by Englert [3]:(1)$$V^{2}+D^{2}\leq 1.$$
*V* is the interference visibility and defined as $\frac{I_{max}-I_{min}}{I_{max}+I_{min}}$, where $I_{max}$ and $I_{min}$ are the maximum and minimum of the output intensity in the interferometer, respectively. *D* is the path distinguishability and reflects the observer’s ability to determine through which path the quanton passes in the interferometer. The equation is derived from the configuration shown in Figure 1a, in which a which-path detector (WPD) system is introduced to a two-path interferometer. According to the quanton’s path of propagation, the state of the WPD system evolves to different corresponding states, and *D* is defined by the distinguishability of these two states. This relation has been experimentally verified for various physical systems [5,6,7,8,9].

As shown in Figure 1b, it was proposed in Ref. [10] that, without the use of a WPD system, the quanton’s path information could also be obtained by an asymmetric beamsplitter (ABS) at the output, which has an arbitrary reflectivity *R* varying from 0 to 1. Two special cases where *R* equals to 0.5 or 1 cause no or full path information to be obtained, and are equivalent to the presence or absence of a symmetric beamsplitter (BS) respectively. Based on this concept, a QBS was considered in Ref. [11], as shown in Figure 1c. A QBS consists of a BS and a two-level controlling system. Depending on the state ($|0{>}_{c}$ or $|1{>}_{c}$) of the controlling system, the BS is in the absence or the presence of the interferometer, and the quanton behaves as a particle or wave accordingly. For simplicity, $|0{>}_{c}$ or $|1{>}_{c}$ are usually denoted as |absence〉 and |presnece〉, which are states of the QBS’s position degree of freedom (DOF). Such a configuration leads to a quantum superposition of particle-state and wave-state [12,13,14,15,16,17,18,19]. Surprisingly, the interference between these two states violates the relation in Equation (Equation 1) [16].

On the other hand, instead of *V* after the output BS, the quanton’s quantum coherence in the $l_{1}$-norm measure [20], *C*, which is defined as the summation of the absolute values of the off-diagonal elements in the quanton’s reduced density matrix before the output BS, is considered to be a candidate to quantify the wave property in the *N*-path interferometer. In Ref. [21], the authors studied the entanglement between the quanton and the WPD system and provided a linear form of the duality relation. As an extension, the authors of Ref. [22] derived a quadratic form, later revising it to a tight form [23]:(2)$${\left(\frac{C+D-\frac{N-2}{N-1}}{\frac{\sqrt{N}}{N-1}}\right)}^{2}+{\left(\frac{C-D}{\sqrt{\frac{N}{N-1}}}\right)}^{2}\leq 1.$$

The relations between the quantum coherence and path distinguishability in Refs. [22,23] have been experimentally demonstrated. See Refs. [24,25,26]. Until now, duality relations based on coherence have been established when wave-like and particle-like properties are classically mixed. However, no studies have dealt with the generalized relation when these two properties are quantum-superimposed. As shown in Ref. [16], quantum superposition adds new phenomena to classical considerations. Thus, an interesting question is whether Equation (Equation 2) can be broken in the quantum case.

To address this question, we propose the addition of a QWPD, which is similar to the QBS and can be in the quantum superposition of the presence and absence on the quanton’s paths, and we study Equation (Equation 2) under such a circumstance. We find that the relation still holds, but the interference between the wave-state and particle-state enables observation of full particle property with the partial presence of the QWPD. Using a pair of polarization-entangled photons, we further experimentally verify our analysis in the two-path case. This study extends the theory of duality relation between coherence and path distinguishability to the quantum case, where these two properties are quantum-superimposed.

## 2. Theory

### 2.1. Model of a Two-Path Interferometer with a QWPD

In this section, we first introduce the principle of a standard (classical) WPD in Figure 1a, and then propose our model of a two-path interferometer with a QWPD and the quantum superposition of the wave-state and particle-state within it. A more general case is presented in Section 2.3.

With the interaction between the quanton and the WPD system, the combined state is expressed as [3]:(3)$${|\psi \rangle }_{qd}=\frac{1}{\sqrt{2}}\left({|0\rangle }_{q}{|d_{0}\rangle }_{d}+{|1\rangle }_{q}{|d_{1}\rangle }_{d}\right),$$
where ${|0\rangle }_{q}$ and ${|1\rangle }_{q}$ are the two path modes of the quanton, and ${|d_{0}\rangle }_{d}$ and ${|d_{1}\rangle }_{d}$ are the two states of the WPD system corresponding to the case where the quanton propagates along the two paths. Two points are noted here. First, the WPD system is not a real facility which is experimentally used to detect the incidence of quantons. Instead, it is rather a physical system which is initialized as ${|d_{ini}\rangle }_{d}$ and coupled to the interferometer. According to the quanton’s propagation (along path 0 or path 1), the WPD system evolves to ${|d_{0}\rangle }_{d}$ or ${|d_{1}\rangle }_{d}$, respectively. Second, ${|d_{0}\rangle }_{d}$ and ${|d_{1}\rangle }_{d}$ are referred by the WPD’s another kind of DOF as a response to the paths, which differs from the position DOF mentioned in Section 1. When ${|d_{0}\rangle }_{d}$ is identical to ${|d_{1}\rangle }_{d}$, we cannot obtain any path information from the state of the WPD system. This leads to maximal coherence and corresponds to the wave-state case. In other cases, some of the path information is revealed. In particular, when ${|d_{0}\rangle }_{d}$ and ${|d_{1}\rangle }_{d}$ are orthogonal to each other, full-path information is revealed, and this corresponds to the particle-state case.

We present our theoretical model in Figure 1d to generate the wave–particle superposition with the QWPD. Since there are no constrictions on ${|d_{ini}\rangle }_{d}$, ${|d_{0}\rangle }_{d}$ and ${|d_{1}\rangle }_{d}$, we temporarily assume that, before interacting with the quanton, ${|d_{ini}\rangle }_{d}$ is identical to ${|d_{0}\rangle }_{d}$. Cases for arbitrary states of the QWPD are considered in Section 2.3.

Following the convention of previous studies on the QBS, we use |presence〉 and |absence〉 to represent the position of the QWPD relative to the interferometer, and they actually represent that the QWPD’s coupling to the interferometer is present and absent, respectively. In the presence case, when the quanton propagates along path 1, it interacts with the QWPD system and transforms its state to ${|d_{1}\rangle }_{d}$. When the quanton propagates along path 0, the state of the QWPD system remains unchanged since it is initialized as ${|d_{0}\rangle }_{d}$. Here, we assume that ${|d_{0}\rangle }_{d}$ and ${|d_{1}\rangle }_{d}$ are orthogonal and represent them as ${|0\rangle }_{d}$ and ${|1\rangle }_{d}$, respectively. In this case, the state of the combined system between the quanton and the QWPD is
(4)$$\frac{1}{\sqrt{2}}\left({|0\rangle }_{q}{|0\rangle }_{d}+{|1\rangle }_{q}{|1\rangle }_{d}\right)|presnece\rangle .$$

We denote $\frac{1}{\sqrt{2}}\left({|0\rangle }_{q}{|0\rangle }_{d}+{|1\rangle }_{q}{|1\rangle }_{d}\right)$ as ${|p\rangle }_{qd}$, which is the particle-state and indicates the particle-like behaviour of the quanton. In the absence case, no interaction occurs between the QWPD system and the quanton, which leads to the state of the combined system as
(5)$$\frac{1}{\sqrt{2}}\left({|0\rangle }_{q}+{|1\rangle }_{q}\right){|0\rangle }_{d}|absence\rangle .$$
Similarly, we denote $\frac{1}{\sqrt{2}}\left({|0\rangle }_{q}+{|1\rangle }_{q}\right){|0\rangle }_{d}$ as ${|w\rangle }_{qd}$, which is the wave-state and indicates the wave-like behaviour of the quanton.

Provided that the position DOF is initially in cosα|presence〉+sinα|absence〉, it is entangled with particle-state and wave-state as
(6)$$\cos\alpha {|p\rangle }_{qd}|presence\rangle +\sin\alpha {|w\rangle }_{qd}|absence\rangle .$$

When this DOF finally collapses on $|+\rangle =\frac{1}{\sqrt{2}}\left(|presence\rangle +|absence\rangle \right)$, the (unnormalized) state of the combined system is
(7)$$(\cos\alpha {|p\rangle }_{qd}+\sin\alpha {|w\rangle }_{qd})|+\rangle ,$$
and we have the quantum superposition between particle-state and wave-state as
(8)$${|\psi \rangle }_{qd}=F\left(\alpha \right)\left(\cos\alpha {|p\rangle }_{qd}+\sin\alpha {|w\rangle }_{qd}\right),$$
where F(α) is the normalization factor as
(9)$$F\left(\alpha \right)={\left(1+\sin\alpha \cos\alpha \right)}^{-\frac{1}{2}}.$$

### 2.2. Duality Relation Based on $l_{1}$-Norm Measure of Quantum Coherence with a QWPD

When N=2, Equation (Equation 2) reduces to $C^{2}+D^{2}\leq 1$. In this section, we study this relation in the interferometer with the QWPD and show that it still holds in such a configuration, but the interference between ${|p\rangle }_{qd}$ and ${|w\rangle }_{qd}$ makes it different from the classical case.

The reduced density matrix of the quanton is $\rho_{q}=Tr_{d}\left(\rho_{qd}\right)$, where $\rho_{qd}={|\psi \rangle }_{qd}\langle \psi |$ is the density matrix for the combined system. Consequently, *C* is calculated to be
(10)$$C\left(\alpha \right)=F^{2}\left(\alpha \right)\left|\sin\alpha (\sin\alpha +\cos\alpha )\right|.$$

The path distinguishbaility D(α) is defined as
(11)D(α)=2P(α)−1
in Refs. [22,23], where P(α) is the maximal success probability of the minimum-error state discrimination (MESD) between the two corresponding states of the detector system. For the calculation of D(α), we first consider P(α). When the quanton is found to propagate along path 0 or 1, the unnormalized states of the QWPD system are
(12)$$\begin{array}{cc} \; & {}_{q}{\langle 0|\psi \rangle }_{qd}=\frac{F(\alpha )}{\sqrt{2}}\left(\cos\alpha +\sin\alpha \right){|0\rangle }_{d}\\ \; & {}_{q}{\langle 1|\psi \rangle }_{qd}=\frac{F(\alpha )}{\sqrt{2}}\left(\sin\alpha {|0\rangle }_{d}+\cos\alpha {|1\rangle }_{d}\right). \end{array}$$

To obtain the quanton’s path information with MESD, we should distinguish the states in the ensemble ${\left\{p_{i},|\tilde{d_{i}}\rangle \right\}}_{i=0}^{1}$, where $|\tilde{d_{i}}\rangle$ is the normalized state of ${}_{q}{\langle i|\psi \rangle }_{qd}$, and $p_{i}={\left|{}_{q}{\langle i|\psi \rangle }_{qd}\right|}^{2}$ is the probability for the QWPD system to be in the state of $|\tilde{d_{i}}\rangle$. They are derived as
(13)$$\begin{array}{cc} \; & p_{0}=\frac{F^{2}\left(\alpha \right)}{2}{\left(\cos\alpha +\sin\alpha \right)}^{2},\;p_{1}=\frac{F^{2}\left(\alpha \right)}{2},\\ \; & {|{\tilde{d}}_{0}\rangle }_{d}={|0\rangle }_{d},\;{|{\tilde{d}}_{1}\rangle }_{d}=\sin\alpha {|0\rangle }_{d}+\cos\alpha {|1\rangle }_{d}. \end{array}$$

According to the relevant theory [27], P(α) is derived as
(14)$$P\left(\alpha \right)=\frac{1}{2}\left(1-\sqrt{1-4p_{0}p_{1}{\left|\langle {\tilde{d}}_{0}|{\tilde{d}}_{1}\rangle \right|}^{2}}\right).$$

From Equations (Equation 10), (Equation 11) and (Equation 14), we have $D\left(\alpha \right)=\sqrt{1-C^{2}\left(\alpha \right)}$, and Equation (Equation 2) still holds when N=2.

Although the duality relation is not broken, the wave–particle superposition still presents new aspects. The interferometer is constructed to be symmetric. However, from Equation (Equation 13), the quanton has unequal probabilities of propagating along the two paths. This is a consequence of the interference between the wave-state and particle-state. To illustrate the relation between the wave-particle interference and the unequal probabilities, we consider the classical mixture between ${|p\rangle }_{qd}$ and ${|w\rangle }_{qd}$ with the density matrix
(15)$$\rho =\cos^{2}\alpha {|p\rangle }_{qd}\langle p|+\sin^{2}\alpha {|w\rangle }_{qd}\langle w|,$$
where the particle-state and the wave-state do not interfere with each other. Similar to the above analysis, the probabilities for the quanton to propagate along path 0 and path 1 are derived as
(16)$$\cos^{2}\alpha {\left|{}_{q}{\langle 0|p\rangle }_{qd}\right|}^{2}+\sin^{2}\alpha {\left|{}_{q}{\langle 0|w\rangle }_{qd}\right|}^{2}=\frac{1}{2}$$
and
(17)$$\cos^{2}\alpha {\left|{}_{q}{\langle 1|p\rangle }_{qd}\right|}^{2}+\sin^{2}\alpha {\left|{}_{q}{\langle 1|w\rangle }_{qd}\right|}^{2}=\frac{1}{2},$$
respectively. Without the wave-particle interference, the quanton still has an equal probability to propagate along the two paths.

As a special case of the unequal probabilities, when $\alpha =\frac{3}{4}\pi$, $p_{1}=1$. Thus, we are sure that the quanton propagates along path 1, and its entire path information is available. Meanwhile, the QWPD system is totally present only when sinα=0. This indicates that full particle-like behavior can be observed when the QWPD system is partially present.

Because the relations based on coherence are declared to be equivalent to Equation (Equation 1) when N=2, we now explain why Equation (Equation 2) holds with a QWPD system, but Equation (Equation 1) is violated with the QBS. In previous relevant studies, *C* was considered to be equivalent to the interference visibility *V* when N=2. Thus, Equation (Equation 2) reduces to Equation (Equation 1) [21]. However, this declaration depends on the condition that the output BS used for calculating *V* is symmetric. In Ref. [16], the quanton’s path information was revealed by the ABS without introducing a WPD system. The asymmetry of the output BS causes the setup to be invalid to illustrate the break of duality based on *C* because this relation is no longer equivalent with the duality relation in Equation (Equation 1). In our study, we consider the quanton’s path information to be revealed by the QWPD system, and the output BS is still symmetric under such circumstances. It is not unusual that, whereas Equation (Equation 1) is broken with a QBS, the duality relation based on coherence with a QWPD still holds.

### 2.3. Duality Relation in a More General Case

In this section, we consider the duality relation in the *N*-path interferometer. The QWPD system is now set to be an *N*-level system with an orthonormal basis {$|d_{0}\rangle$, $|d_{1}\rangle$, ⋯, $|d_{N-1}\rangle$}, each corresponding to one of the *N* paths. The initial state of the QWPD system is generally expressed as ${|d_{ini}\rangle }_{d}=\sum_{i=0}^{N-1}\beta_{i}|d_{i}\rangle ,\beta_{i}\in R$. We draw the same conclusion as that derived in Section 3, i.e., Equation (Equation 2) is not violated, and full particle information is available with the partial presence of the QWPD system.

Similar to Equation (Equation 8), the superposition of the wave-state and particle-state is expressed as
(18)$${|\Psi \rangle }_{qd}=F_{N}\left(\alpha \right)\left(\cos\alpha {|P\rangle }_{qd}+\sin\alpha {|W\rangle }_{qd}\right),$$
where ${|P\rangle }_{qd}$ and ${|W\rangle }_{qd}$ are the particle-state and wave-state as
(19)$${|P\rangle }_{qd}=\sum_{i=0}^{N-1}{|i\rangle }_{q}{|d_{i}\rangle }_{d},\;{|W\rangle }_{qd}=\sum_{i=0}^{N-1}{|i\rangle }_{q}{|d_{ini}\rangle }_{d},$$
and $F_{N}\left(\alpha \right)$ is the normalization factor as
(20)$$F_{N}\left(\alpha \right)={\left(N+\sin\alpha \cos\alpha \left(\langle P|W\rangle +\langle W|P\rangle \right)\right)}^{-\frac{1}{2}}.$$

Here, ${|P\rangle }_{qd}$ and ${|W\rangle }_{qd}$ are unnormalized for convenience, and the normalization factor is absorbed into $F_{N}\left(\alpha \right)$. Equation (Equation 19) can be rewritten as
(21)$${|\Psi \rangle }_{qd}=\sum_{i=0}^{N-1}\sqrt{p_{i}}{|i\rangle }_{q}{|\hat{d_{i}}\rangle }_{d},$$
where ${|\hat{d_{i}}\rangle }_{d}\equiv {|{\tilde{d}}_{i}\rangle }_{d}{{|}_{d}{\langle {\tilde{d}}_{i}|{\tilde{d}}_{i}\rangle }_{d}|}^{-\frac{1}{2}}$ with ${|{\tilde{d}}_{i}\rangle }_{d}\equiv \cos\alpha {|d_{i}\rangle }_{d}+\sin\alpha {|d_{ini}\rangle }_{d}$, and $p_{i}\equiv F_{N}^{2}\left(\alpha \right)\left(1+\beta_{i}\sin2\alpha \right)$ with $\beta_{i}=\langle d_{i}|d\rangle$ is the probability that a quanton propagates along the *i*th path. Because Equation (Equation 21) is the standard form of the entangled state between the quanton and the WPD system, the relation in Equation (Equation 2) is not violated. However, as we show next, full information from the particle property is still attainable with the partial presence of the QWPD system.

Based on Equation (Equation 21), the density matrix of the combined system is
(22)$$\rho_{qd}=\sum_{i,j=0}^{N-1}\sqrt{p_{i}p_{j}}{|i\rangle }_{q}\langle j|⊗{|\hat{d_{i}}\rangle }_{d}\langle \hat{d_{j}}|,$$
and the element ${\left(\rho_{q}\right)}_{i,j}$ in the reduced density matrix of the quanton is
(23)$$\begin{array}{cc} \; & \sqrt{p_{i}p_{j}}\sum_{k=0}^{N-1}\langle d_{k}|\hat{d_{i}}\rangle \langle \hat{d_{j}}|d_{k}\rangle \\ \; & =\sqrt{p_{i}p_{j}}{\left|\langle {\tilde{d}}_{i}|{\tilde{d}}_{i}\rangle \right|}^{-\frac{1}{2}}{\left|\langle {\tilde{d}}_{j}|{\tilde{d}}_{j}\rangle \right|}^{-\frac{1}{2}}\sum_{k=0}^{N-1}\langle d_{k}|\tilde{d_{i}}\rangle \langle \tilde{d_{j}}|d_{k}\rangle . \end{array}$$

We focus on the off-diagonal elements (i.e., i≠j), and the summation in Equation (Equation 23) is simplified as
(24)$$\sum_{k=0}^{N-1}\langle d_{k}|\tilde{d_{i}}\rangle \langle \tilde{d_{j}}|d_{k}\rangle =\sin^{2}\alpha +\sin\alpha \cos\alpha \left(\beta_{i}+\beta_{j}\right).$$

In the two-path case of Section 3, we assume ${|d_{ini}\rangle }_{d}={|d_{0}\rangle }_{d}$, and the full particle property was obtained when the quanton was determined to propagate along path 1 (i.e., $p_{1}=1$). This is impossible in the *N*-path case. However, when ${|d_{ini}\rangle }_{d}=\frac{1}{\sqrt{N}}\sum_{i=0}^{N-1}{|d_{i}\rangle }_{d}$, and $\alpha =\pi -\arctan\frac{2}{\sqrt{N}}$, all off-diagonal elements vanish, $\rho_{q}$ is totally de-coherent, and the quanton remains in a particle-state. Under such circumstances, $p_{i}=\frac{1}{N}\;\forall i$, and $\langle \tilde{d_{i}}|\tilde{d_{j}}\rangle =0\;\forall i\neq j$. The states of the QWPD system are completely distinguishable.

## 3. Experiment and Results

### 3.1. Principle of the Experimental Design

We further design a scheme to experimentally verify our theoretical analysis when N=2. Instead of focusing on the interaction between the quanton and the QWPD system shown in Figure 1d, we directly generate the quantum superposition between wave-state and particle-state in Equation (Equation 8), which is analogous to the methods used in previous experimental verifications [24,25,26] of duality relations with a standard WPD system.

In experiments, we generate a pair of polarization-entangled photons which represent the quanton and the QWPD system, respectively, and use polarization DOF of each photon to encode the path modes of the quanton and the response of the QWPD system. To make it more readable, before the introduction of the experimental setup, we first elaborate how to generate the states in Equations (Equation 6) and (Equation 8) with bipartite entanglement. Two conditions are supposed to be met for our purposes.
For the particle-state ${|p\rangle }_{qd}$ in Equation (Equation 6), the entanglement should be maintained, while, for the wave-state ${|w\rangle }_{qd}$ in Equation (Equation 6), the entanglement should be eliminated.An ancillary DOF is required to represent the position DOF in Equation (Equation 6), i.e., |presence〉 and |absence〉. It should first be entangled with ${|p\rangle }_{qd}$ and ${|w\rangle }_{qd}$, and then collapse on |+〉 to generate the state in Equation (Equation 8).

To meet these two conditions, we design our experimental setup shown in Figure 2a. A pair of polarization-entangled photons at 810 nm is generated via a type-I spontaneous parametric down-conversion process [28] in two cascaded 600-μm-thick BBO crystals, pumped by a 100-mW continuous beam at 405 nm. These two photons representing the quanton and the QWPD system are entangled as
(25)$$|\Phi \rangle =\frac{1}{\sqrt{2}}\left({|H\rangle }_{q}{|H\rangle }_{d}-{|V\rangle }_{q}{|V\rangle }_{d}\right),$$
where |H〉 and |V〉 are the horizontal and vertical polarization states of the photons. The polarizations of the two photons are used to encode ${|p\rangle }_{qd}$ and ${|w\rangle }_{qd}$, while the transmission and reflection of the quanton photon by the polarization-dependent beamsplitter (PBS) are used to encode the position DOF, and are controlled by the angle of HWP2 in Figure 2a. Though the QWPD’s position DOF is represented by the path mode of the quanton photon, this method is sensible because the path mode DOF is independent of the polarization DOFs of the two photons. We now explain how the setup works to meet the two conditions.

#### 3.1.1. Generation of ${|p\rangle }_{qd}$ and ${|w\rangle }_{qd}$

First, we explain the realization of condition 1 with simplified Figure 2b,c, which represent the implementations of ${|p\rangle }_{qd}$ and ${|w\rangle }_{qd}$, respectively. In Figure 2a, the HWP1, HWP3, and HWP3${}^{\prime }$ are set as $\frac{\pi }{4}$, $\frac{\pi }{4}$ and $\frac{3}{8}\pi$. When the HWP2 is set as 0, the quanton photon’s propagation is shown in Figure 2b.

From the state in Equation (Equation 25), when the quanton photon passes through a beam-displacer (BD), its path modes of horizontal and vertical polarizations are spatially separated by 3 mm. After the first HWP1 placed on the lower path, the polarizations of both paths are horizontal. After HWP2 placed on both paths, the quanton photon is transmitted by the PBS. Finally, after HWP3 placed on the upper path, these two paths are recombined, and the state of the combined system is given by
(26)$$\frac{1}{\sqrt{2}}\left({|V\rangle }_{q}{|H\rangle }_{d}-{|H\rangle }_{q}{|V\rangle }_{d}\right)|transmitted\rangle ={|p\rangle }_{qd}|transmitted\rangle .$$

The bipartite entanglement is maintained for the particle-state. To make Equation (Equation 26) consistent with Equation (Equation 4), we encode ${|V\rangle }_{q}$, $-{|H\rangle }_{q}$, ${|H\rangle }_{d}$, ${|V\rangle }_{d}$ and |transmitted〉 as ${|0\rangle }_{q}$, ${|1\rangle }_{q}$, ${|0\rangle }_{d}$, ${|1\rangle }_{d}$, and |presence〉, respectively. When the QWPD photon’s polarization is ${|H\rangle }_{d}$ (or ${|V\rangle }_{d}$), we can get that the quanton photon’s polarization is ${|V\rangle }_{q}$ (or $-{|H\rangle }_{q}$). Knowledge of the quanton’s information is revealed by the QWPD photon, and the quanton photon behaves as a particle.

Similarly, when the HWP2 in Figure 2a is set as $\frac{\pi }{4}$, the quanton photon’s propagation is shown in Figure 2c. After the HWP2, the polarizations of both paths are vertical, and the quanton photon is reflected by the PBS. If we block the right path and collect the quanton photon only on the left path, the state of the two photon polarizations is ${|V\rangle }_{q}{|H\rangle }_{d}$, and the bipartite entanglement is eliminated. After a third HWP$3^{\prime }$ having the angle of $\frac{3}{8}\pi$, the state of the combined system is
(27)$$\frac{1}{\sqrt{2}}\left({|V\rangle }_{q}-{|H\rangle }_{q}\right){|H\rangle }_{d}|reflected\rangle ={|w\rangle }_{qd}|reflected\rangle .$$

With the above encoding and |reflected〉 representing |absence〉, Equation (Equation 27) is consistent with Equation (Equation 5). Since the polarization of the QWPD photon is independent of that of the quanton photon, no knowledge of the quanton’s information is revealed, and the quanton behaves as a wave. Thus, the maintaining and elimination of the bipartite entanglement for the particle case and wave case, respectively, are realized.

#### 3.1.2. Generation of Quantum Superposition between ${|p\rangle }_{qd}$ and ${|w\rangle }_{qd}$

Second, we explain how |transmitted〉 and |reflected〉 works as the position DOF to meet condition 2. According to the explanation for condition 1, when the HWP2 in Figure 2a is set as an arbitrary angle α, the propagation of the quanton photon is in the superposition between Figure 2b,c, This is shown as the superposition between the clockwise red lines and the counterclockwise blue lines inside the displaced Sagnac loop [29] in Figure 2a. Under such a circumstance, |transmitted〉 and |reflected〉 are entangled with ${|p\rangle }_{qd}$ and ${|w\rangle }_{qd}$ as
(28)$$\sqrt{2}\cos\alpha {|p\rangle }_{qd}|transmitted\rangle +\sin\alpha {|w\rangle }_{qd}|reflected\rangle .$$

The collapse on |+〉 is implemented by a non-polarizing BS (NPBS) to recombine |transmitted〉 and |reflected〉 at the output, and the (unnormalized) state evolves to
(29)$$(\sqrt{2}\cos\alpha {|p\rangle }_{qd}+\sin\alpha {|w\rangle }_{qd})|+\rangle .$$

Corresponding to Equation (Equation 8), we generate a wave–particle superposition as
(30)$$F^{\prime }\left(\alpha \right)\left(\sqrt{2}\cos\alpha {|p\rangle }_{qd}+\sin\alpha {|w\rangle }_{qd}\right),$$
where $F^{\prime }\left(\alpha \right)$ is the normalization factor. Because in the generation of ${|w\rangle }_{qd}$ the quanton has a probability of $\frac{1}{2}$ to be absorbed by the trap, Equation (Equation 30) differs from Equation (Equation 8) with the coefficients of ${|p\rangle }_{qd}$. However, Equation (Equation 30) is equivalent to Equation (Equation 8) in that it also generates a continuous quantum superposition between ${|p\rangle }_{qd}$ and ${|w\rangle }_{qd}$.

The setup is arranged to ensure that the three path modes interfere stably. First, each mode propagates through the same elements or an equal number of waveplates. Second, the PBS and NPBS are glued together as a cube.

### 3.2. Measuring Coherence and Path Distinguishability

In the experiment, we perform single-qubit QST [30] on the quanton photon, and we calculate its coherence, *C*, according to the measured density matrix. Path distinguishability *D* is obtained by projective measurement on the QWPD photon as follows.

Because the amplitudes in Equation (Equation 30) differ from those in Equation (Equation 8), the QWPD states and their probabilities in Equation (Equation 13) are calculated as
(31)$$\begin{array}{cc} \; & {|{\tilde{d}}_{0}\rangle }_{d}={|H\rangle }_{d},\;{|{\tilde{d}}_{1}\rangle }_{d}=\frac{1}{\sqrt{{\bar{p}}_{1}}}\left(\cos\alpha {|V\rangle }_{d}+\sin\alpha /\sqrt{2}{|H\rangle }_{d}\right)\\ \; & p_{0}=\frac{{\bar{p}}_{0}}{{\bar{p}}_{0}+{\bar{p}}_{1}},p_{1}=\frac{{\bar{p}}_{1}}{{\bar{p}}_{0}+{\bar{p}}_{1}}, \end{array}$$
where
(32)$${\bar{p}}_{0}={\left(\cos\alpha +\frac{1}{\sqrt{2}}\sin\alpha \right)}^{2},{\bar{p}}_{1}=\cos^{2}\alpha +\frac{1}{2}\sin^{2}\alpha .$$

According to Ref. [27], in order to achieve the maximal success probability in discrimination between states ${|{\tilde{d}}_{0}\rangle }_{d}$ and ${|{\tilde{d}}_{1}\rangle }_{d}$, the projectors in the positive operator-valued measure are
(33)$$\Pi_{0}=|\pi_{0}\rangle \langle \pi_{0}|,\Pi_{1}=|\pi_{1}\rangle \langle \pi_{1}|=I-\Pi_{0},$$
with
(34)$$|\pi_{0}\rangle =\cos\delta |0^{\prime }\rangle +\sin\delta |1^{\prime }\rangle .$$

δ and the basis are derived by
(35)$$\begin{array}{cc} \; & \tan\delta =\frac{\lambda -\left(p_{0}-p_{1}\right)\cos^{2}\gamma }{\sin\gamma \cos\gamma }\\ \; & \lambda =\frac{1}{2}\left(p_{0}-p_{1}+\sqrt{1-4p_{0}p_{1}\cos^{2}2\gamma }\right)\\ \; & \cos^{2}\gamma =\frac{1}{2}\left(1+\langle \tilde{d_{0}}|\tilde{d_{1}}\rangle \right) \end{array}$$
and
(36)$$\begin{array}{cc} \; & |0^{\prime }\rangle =\frac{1}{2\cos\gamma }\left(|\tilde{d_{0}}\rangle +|\tilde{d_{1}}\rangle \right)\\ \; & |1^{\prime }\rangle =\frac{1}{2\sin\gamma }\left(|\tilde{d_{0}}\rangle -|\tilde{d_{1}}\rangle \right). \end{array}$$
P(α) in Equation (Equation 14) is expressed as
(37)$$P\left(\alpha \right)=p_{0}\langle \pi_{0}|{\tilde{d}}_{0}|\pi_{0}\rangle +p_{1}\langle \pi_{1}|{\tilde{d}}_{1}|\pi_{1}\rangle .$$

For each value of α, we first calculate $|\pi_{0}\rangle$ and $|\pi_{1}\rangle$ according to the equations from Equations (Equation 31) to (Equation 36). Then, we record the four coincidence counts, $n_{V0}$, $n_{V1}$, $n_{H0}$, and $n_{H1}$, where $n_{V0}$ is the count when the polarizers in front of the quanton photon and the QWPD photon are set as |V〉 and $|\pi_{0}\rangle$, respectively, and the other three counts have similar meanings. P(α) is calculated as
(38)$$P\left(\alpha \right)=\frac{n_{V0}+n_{H1}}{n_{V0}+n_{V1}+n_{H0}+n_{H1}}.$$

Then, D(α) is obtained by Equation (Equation 11).

Our measured coherence and path distinguishability are shown in Figure 3. From Figure 3a,b, the measured coherence and path distinguishability are in agreement with the theoretical analysis. When α=0, the quanton exhibits particle-like behavior, and, as α increases, it is gradually morphed to a wave state with an increase in C(α) and a decrease in D(α). When α reaches $\frac{\pi }{2}$, the quanton is in the full-wave state. When α varies from $\frac{\pi }{2}$ to π, C(α) does not gradually decrease to 0. Instead, it drops to 0 fast and remains at a low-value region. On the contrary, D(α) rises to 1 fast and stays at a high-value region. This is the consequence of the interference between the wave-state and particle-state when they are in a quantum superposition. In particular, when $\alpha =\pi -\arctan(\sqrt{2})$, D(α) reaches the maximum of one theoretically, and we have full-path information when the QWPD system is partially present. More experimental details are exhibited in the Appendix A.

The deviation of the measured results from the theoretical analysis is caused by the inaccuracy in the rotation of the waveplates and polarizers, the detection loss of the photons, the imperfect interference of the path modes, the additional phase introduced when rotating the HWPs, and, mainly, the imperfect entanglement between the two photons caused by the imperfect interference in the two BDs, which is obvious when D(α) is relatively large. In Figure 3c, we exhibit the plot of (C,D) for each α, and the theoretical result is described as $C{\left(\alpha \right)}^{2}+D{\left(\alpha \right)}^{2}=1$. When D(α) is relatively large, the measured (C,D) lies inside the circle. This is because, when the two photons are not perfectly entangled, the states of the QWPD corresponding to ${|V\rangle }_{q}$ and ${|H\rangle }_{q}$ are mixed states, and, according to Refs. [22,23], $C^{2}+D^{2}$ is smaller than one. We measured the *S* value in the CHSH inequality [31] when α=0, and the value was 2.5, which is not close to the theoretical maximum of $2\sqrt{2}$, implying the imperfect entanglement.

## 4. Conclusions

In conclusion, we studied the duality relation based on quantum coherence in the $l_{1}$-norm measure, when there was superposition of the presence and absence of a QWPD, which leads to the wave–particle superposition. In contrast with previous similar studies, the relation still holds under such a condition, but the interference between wave-state and particle-state facilitates the obtainment of the quanton’s full-path information when the QWPD system is partially present. Using a pair of polarization-entangled photons, we experimentally generated the wave–particle superposition and verified our theoretical analysis using the two-path case. One of the limitations of this work is that the demonstration was only performed when N=2, and the experiment for the multi-path interference was expected. This study extends the duality relation to the quantum case and provides a new aspect of duality relation with quantum wave-particle superposition.

## Figures and Tables

**Figure 1 entropy-23-00122-f001:**
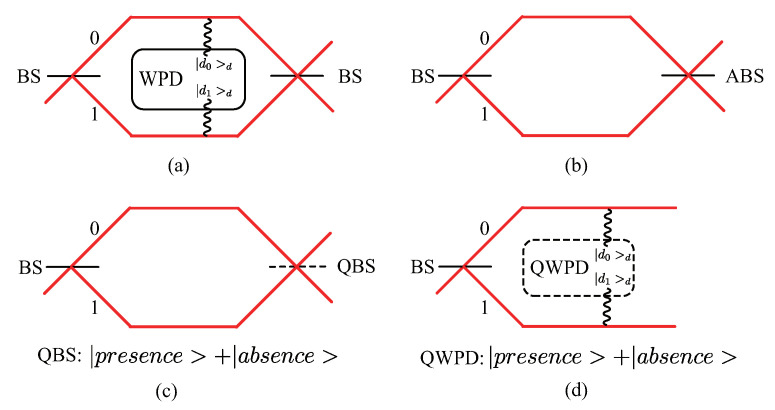
Models used to study the duality relation. (**a**) original model for deriving the duality relation. The WPD system is introduced in the two-path interferometer. $|d_{0}\rangle$ ($|d_{1}\rangle$) is the state of the WPD system’s response when the quanton propagates along path 0 (1), and it reveals the quanton’s path information. (**b**) In the absence of the WPD system, we can obtain the quanton’s path information from the ABS. (**c**) QBS enables wave-particle superposition. The dashed line indicates the superposition between presence and absence, and so does the dashed line in (**d**). (**d**) Our model used to study duality relation based on the quantum coherence *C* with a QWPD, which similarly leads to wave–particle superposition. We omit the output BS because we only care about the state before it.

**Figure 2 entropy-23-00122-f002:**
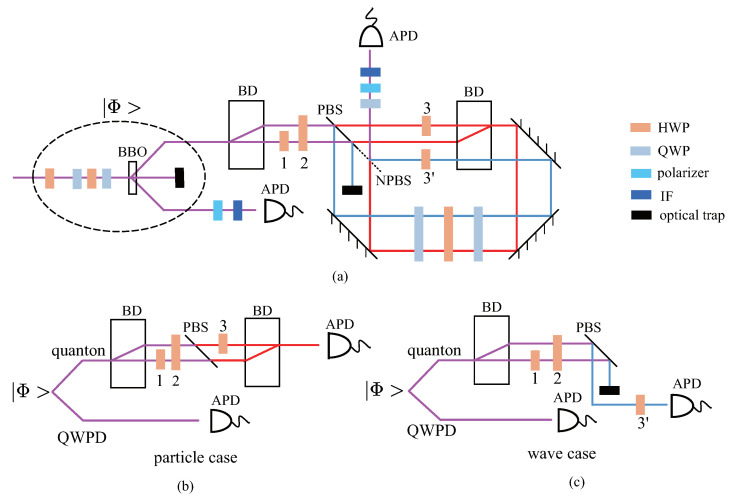
Experimental setup to generate wave–particle superposition. (**a**) the whole setup. The pump beam is initialized as $\frac{1}{\sqrt{2}}\left(|H\rangle -|V\rangle \right)$ using a half-wave plate (HWP), generating a pair of polarization-entangled photons in a pair of β-Ba_2_B_2_O_4_ (BBO) crystals. HWP1, HWP2, HWP3, and HWP3’ are set as $\frac{\pi }{4}$, $\frac{\alpha }{2}$, $\frac{\pi }{4}$, and $\frac{3}{8}\pi$, respectively. Before the two photons are coupled into fibers, they pass through two interference filters (IFs) having bandwidths of 3 nm. Then, they are counted by two avalanche photodiode detectors (APDs), and the signals are processed in a coincidence detection with a time window of 5 ns. The two sandwich-like configurations on the pump beam and the quanton photon, consisting of two quarter-wave plates (QWPs) and an HWP, are used to adjust the relative phase between horizontal and vertical polarizations. The QWP on the quanton photon is used only in the quantum state tomography (QST). (**b**,**c**) are simplified schemes to illustrate the generation of particle-state and wave-state in (**a**). More details are in the main text.

**Figure 3 entropy-23-00122-f003:**
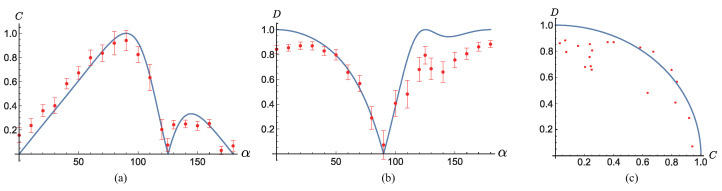
Experimental results of coherence *C* (**a**), path distinguishability *D* (**b**), and the duality relation between them (**c**). We vary α from 0${}^{\circ }$ to 180${}^{\circ }$ with an interval of 10${}^{\circ }$. Moreover, we measure *C* and *D* when α is approximately 125${}^{\circ }$, which corresponds to the case where the quanton’s full path information is obtained with the QWPD’s partial presence. The measured results are shown with red dots, and the theoretical comparisons are shown with blue curves. The error bars are calculated using Poissonian statistics generated based on experimental measured counts.

## Data Availability

The data presented in this study are available in the Appendix A.

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
