# Peer review of "Wave-Particle Duality Relation with a Quantum Which-Path Detector"

_entropy, 2021, doi:10.3390/e23010122_

Round 1
Reviewer 1 Report
This work studies wave-particle duality via a two-path photon interferometer. Instead of considering the Mach-Zehnder scenario where interference visibility (V) is used as a probe of wave-particle nature, the authors consider a coherence measure (C) using which-path detectors (WPDs). The novelty comes from considering quantum-WPDs (QWPD), which can operate in a coherent superposition of monitoring the path or not, to assess the effect on a Coherence-(path) Distinguishability (D) relation.
As the authors point out, a V-D relation has been studied using a Mach-Zehnder interferometer with a quantum output beamsplitter. This work studies a C-D relation using a quantum WPD, finding no change from the case of a classical mixture, although entanglement between the particle and the QWPD does allow observation of particle-like behaviour without a perfect measurement by the QWPD. This is also verified experimentally. I consider this of suitable novelty for Entropy, after the following points are addressed and clafiried.
Introduction: The introduction is ok, but very heavy on acronyms and terms like visibility, coherence and distinguishability, before these terms have been defined. As I say, I think it’s acceptable, but I’m sure the authors could make the introduction more readable.
Figure 1: Although it is defined in the text, I think the authors should define “C” as a coherence measure in the caption to this figure.
Line 27: It took me a long time to understand that the authors were referring to a superposition of the measuring device being present and absent. “Presence and absence states” is ambiguous, perhaps referring to the particle. I think the authors should explain this carefully and clearly, even if it seems obvious to them.
Line 45: The authors start this section stating that they will consider “a two-path interferometer with a QWPD…” but in the next line I believe they discuss a standard WPD system (Equation (3) ). This is confusing, include a statement like “at first we consider a standard WPD scenario” or similar.
Line 53: Something else I found confusing was that in e.g. Fig. 1(d) a QWPD is drawn in each arm of the interferometer. If I understand, this isn’t correct, the detector is only sensitive to the presence of a photon in one arm of the interferometer, it measures 0 or 1 in a single arm, and this is the entire which-way measurement. Fig 1(d) made the rest of the discussion confusing to me, since I was expecting two physical positions of the detector with two possible measurement outcomes each.
Equation (6): I do not believe this equation is necessary, or that it makes much sense. How is |presence>, |absence> separate from the description in equations (4) and (5)? I think (6) may be incorrect, it’s certainly unnecessary. Equation (7) is correct, however, but doesn’t really follow from (6).
Just after Equation (9): D is not defined in Equation (2), in contrast to what the text states.
Equation (11): The meaning of the terms in this equation are very unclear to me, especially … is it just a normalized version of equation (10)?
Equation (12): What is P(α), it is not defined.
Line 65: You state in a couple of places in the paper the phrase “This is a consequence of the interference between the wave and the particle states”, but you don’t formally justify it. Could it not be a consequence of the fact the quanton and the QWPD are entangled?
Line 67: I think you mean sin(α) =0.
Equations (14-16): FN(α) and βi aren’t defined.
Line 97: Since it is key, I think you need to describe and discuss in detail in this introduction how one of the entangled photons acts as a QWPD. I understand the mathematical demonstration that follows, but at this point I think it needs an intuitive discussion, rather than simply a statement.
Author Response
Thanks for the comment. Please see the attachment of our responses. For convenience, we have also provided the revised manuscript after the response. The revisions are colored, and some of them are made according to other referees' comments.

Reviewer 2 Report
This paper studies wave-particle duality relations in which a beam splitter has an extra degree of freedom, allowing it to be in a superposition of being present or absent. The first part of the paper is pretty clear, though I would recommend one small change. In section 2.1 p is used to denote the state of the particle when the beam splitter is present and w to denote its state when the beam splitter is absent. In section 2.3 P is used to denote the particle state when the beam splitter (actually, a multiport) is absent and W to denote the state when it is present. It would be a good idea to keep the use of p and w consistent throughout the paper.
I had problems with section 3. It is easiest to discuss them by making reference to Figure 2. Figure 2a shows how to prepare a beam-splitter present state (the analog of |p> in Eq. 4) while 2b shows how to prepare the beam-splitter-absent state (the analog of |w> in Eq. 5). My first problem is with Fig. 2b. The preparation of the beam-splitter-absent state is conditional; the photon has to not be absorbed by the trap, and this will only occur with some probability. How does this conditional preparation affect the analysis in section 2? Maybe there is essentially no effect, but some further explanation would be a good idea. My second problem is the following. The arrangements in Figures 2a and 2b are combined in Figure 2c. However, Figures 2a and 2b require different sets of half-wave plates, yet in figure 2c, there is just one set of half-wave plates. How are the states produces in Figure 2c then related to those produces in Figures 2a and 2b?
In conclusion, the paper would significantly benefit from a more detailed explanation of how the experimental set up works, and an expanded explanation of how the experiment is connected to the theory in section 2.
Author Response
Thanks for the comments. Please see the attachment of our responses. For convienience, we have provided the revised manuscript after the response. The revisions are colored, and some of them are made according to other referees' comments.
